# Water-Gas Two-Phase Flow Behavior of Multi-Fractured Horizontal Wells in Shale Gas Reservoirs

**Lei Li [1,2]** , **Guanglong Sheng [3,\*]** **and Yuliang Su [1,2]**

1 Key Laboratory of Unconventional Oil & Gas Development, China University of Petroleum (East China), Ministry of Education, Qingdao 266580, China; upclilei@outlook.com (L.L.); suyuliang@upc.edu.cn (Y.S.)
2 School of Petroleum Engineering, China University of Petroleum (East China), Qingdao 266580, China
3 School of Petroleum Engineering, Yangtze University, Wuhan 430100, China
\* Correspondence: shenggl2019@yangtzeu.edu.cn

**Abstract:** Hydraulic fracturing is a necessary method to develop shale gas reservoirs effectively and economically. However, the flow behavior in multi-porosity fractured reservoirs is difficult to characterize by conventional methods. In this paper, combined with apparent porosity/permeability model of organic matter, inorganic matter and induced fractures, considering the water film in unstimulated reservoir volume (USRV) region water and bulk water in effectively stimulated reservoir volume (ESRV) region, a multi-media water-gas two-phase flow model was established. The finite difference is used to solve the model and the water-gas two-phase flow behavior of multi-fractured horizontal wells is obtained. Mass transfer between different-scale media, the effects of pore pressure on reservoirs and fluid properties at different production stages were considered in this model. The influence of the dynamic reservoir physical parameters on flow behavior and gas production in multi-fractured horizontal wells is studied. The results show that the properties of the total organic content (TOC) and the inherent porosity of the organic matter affect gas production after 40 days. With the gradual increase of production time, the gas production rate decreases rapidly compared with the water production rate, and the gas saturation in the inorganic matter of the ESRV region gradually decreases. The ignorance of stress sensitivity would cause the gas production increase, and the ignorance of organic matter shrinkage decrease the gas production gradually. The water film mainly affects gas production after 100 days, while the bulk water has a greater impact on gas production throughout the whole period. The research provides a new method to accurately describe the two-phase fluid flow behavior in different scale media of fractured shale gas reservoirs.

**Keywords:** flow behavior; water-gas two-phase flow; multi-porosity; shale gas reservoirs; fracturing horizontal wells

---

## 1. Introduction

Fluid flow in shale gas reservoirs is affected by multiple migration mechanisms [1,2], static structure characteristics of porous media [3,4] and dynamic changes of pore size [5,6]. Fluid flow in the pores above the micron scale of the shale reservoir can be described by the traditional Darcy's law, but fluid flow in the nanopore can no longer be characterized by the traditional Darcy's law [7,8]. Shale gas exists in the form of free gas and adsorbed gas in the reservoir [2]. The migration mechanism is complex in different scale pores (slip flow, adsorption and desorption, Knudsen diffusion, surface diffusion, etc.), which is affected by temperature, pressure and wall properties [9–14]. Besides, the porous media in shale reservoirs have specific heterogeneity, the pore size distribution, pore geometry, tortuosity and surface roughness have a great impact on fluid migration [15–17]. As studied, the pore systems in

shale reservoirs are multiple-scale with nano- and micro-meter pores [18,19], and the micropores play important effects on geometric, topological and transport properties of the pore systems [20]. Scholars also have conducted a lot of research on the pore change during depressurization of shale reservoirs, including stress sensitivity [21–23], organic matter shrinkage [5,6,24] and adsorption layer changes [25]. Pore size shrinkage caused by pore stress sensitivity, pore size expansion caused by organic matter shrinkage and free gas expansion caused by a thickness change of the adsorption layer has attracted attention [24].

Since shale gas reservoirs have very low permeability and natural fractures that do not fully connect with each other, it is difficult to get economic production [26,27]. Therefore, horizontal wells and hydraulic fracturing techniques have been widely adopted in order to maximize the flow contact area and efficiently produce shale gas [28]. As reported by United States Energy Information Administration (EIA) in 2019, unconventional reservoirs can cost-effectively produce gas only by using a special stimulation technique, like hydraulic fracturing, or other special recovery process and technology [29]. Hydraulic fracturing not only produces hydraulic fractures with high conductivity, but also creates complex induced fracture around hydraulic fractures [28,30,31]. The area containing the complex induced fracture is called the effectively stimulated reservoir volume (ESRV), which greatly enhances the reservoir permeability and has a decisive influence on production well production [32,33].

Recently, the apparent permeability mode has been widely used to describe the various migration mechanisms of gas and liquid flow in micro-nano pores, including Knudsen diffusion, surface diffusion, viscous flow, slip flow, etc. The apparent permeability models mainly include Javadpour's model based on pore size, Civan's model based on Knudsen number and DGM model based on diffusion coefficient [24]. At the same time, the apparent porosity model of organic/inorganic matter is proposed to describe the gas state in micro-nano pores [7]. According to the state of gas occurrence, the pore volume of organic/inorganic matter can be divided into two parts: Pore volume filled with free gas (free gas porosity) and pore volume filled with adsorbed gas (adsorbed gas porosity). Gas concentrations in different parts differ greatly, so the inherent porosity of porous media in shale cannot be used to directly estimate geological reserves. Therefore, the apparent porosity is proposed by scholars. Through the apparent porosity, the gas reserves and the residual amount of free gas and adsorbed gas at different pore pressures can be directly estimated [7]. Based on the apparent porosity/permeability model, conventional numerical simulation methods can be used to describe the flow mechanism of organic and inorganic matter in shale gas reservoirs.

The distribution of induced fracture spacing is generally too complex to give a quantitative representation [34,35]. The induced fracture spacing distribution commonly used in geology is mainly in the form of uniform, linear, normal and exponential functions [36,37]. Belani and Jalali proposed a more general fracture-matrix flow model in which the induced fracture spacing can be described by uniform and normal functions [38]. Johns and Jalali proposed a comprehensive analytical model to study the pressure transient behavior of natural fracture reservoirs with exponential and linear distribution of induced fracture spacing [39]. Spivey and Lee studied the effect of exponential distribution of induced fracture spacing on cross-flow coefficients and pressure distribution [40]. Those scholars proposed a fractal diffusion equation (FDE) describing the distribution of natural fractures. The distribution of induced fracture spacing accords with the fractal scale feature, which defines the fractal dimension of natural fractures to characterize the equivalent porosity/permeability of the fracture system in multiple-porosity medium [41–44]. Sheng et al. proposed the distribution of fractal induced-fracture spacing in radial and Cartesian coordinate systems [45]. A new method to calculate the tortuosity index of induced fracture is proposed, and the fractal dimension of induced-fracture aperture (dfa) is proposed to describe the distribution of fractal induced-fracture aperture. The fractal fracture porosity/permeability models are proposed by combining fractal induced fracture spacing and fractal induced fracture aperture. The new method can describe the induced fracture accurately.

Shale gas reservoirs are generally developed using multi-fractured horizontal wells. Numerical simulations and analytical/semi-analytical models are often used to describe the flow behavior of

multi-fractured horizontal wells [46,47]. The analytical/semi-analytical method describes the seepage of two-phase in reservoirs and fractures by solving partial differential equations [48]. The linear flow model is one of the commonly used analytical/semi-analytical models for multi-fractured horizontal wells, including bilinear flow models [49], trilinear flow models [50], four-region composite linear flow models [51] and five-region composite linear flow models [52]. The method is simple and fast, but it cannot characterize the complex flow behavior in organic matter, inorganic matter and induced fractures in the shale gas reservoir as described above. Numerical simulation models can handle more complex fracture networks, but the method is computationally time-consuming, requiring high expertise when building models [46]. However, this method can characterize the influence of pressure changes on reservoir physical parameters at different production moments, and can accurately characterize the fluid migration mechanism of shale gas reservoirs.

In this paper, the complex migration mechanisms of fluids in the micro-scale pores of organic and inorganic media, the static structural characteristics of porous media, the dynamic changes of pore size, and the distribution of induced fractures spacing/aperture of shale gas reservoirs are comprehensively considered. A multi-media water-gas two-phase flow model of multi-fractured horizontal wells is established. Based on the model proposed in this paper, the dynamic properties of shale gas reservoirs were considered in flow simulation, including the dynamic porosity/permeability of organic and inorganic matter due to stress sensitivity and organic matter shrinkage. Besides, the complex water distribution in different reservoir volume (ultra-low water saturation in unstimulated reservoir volume (USRV) and high water saturation in ESRV) and its influence on flow behavior (single gas flow in pores with ultra-low water saturation and water-gas flow in pores with high water saturation) can also considered in the model.

## 2. Flow Mechanisms of Gas and Water in Multi-Porosity Media

### 2.1. Flow Mechanism in the Porosity of Organic Matter

The organic matter in shale reservoirs is characterized by hydrophobicity. The gas in porous media is diverse (free gas, adsorbed gas), and the pore structure is complex (pore cross-section, pore size distribution, pore tortuosity and surface roughness). In the process of depressurization, the pores are affected by multiple effects (pore enhancement due to organic matter shrinkage, pore shrinkage due to stress sensitivity and thickness thinning of the adsorption layer). At the same time, the micro-scale pores have a large specific surface area, and the effect of surface effect on fluid migration has been gradually greater than inertial force, exhibiting complex flow characteristics.

The apparent porosity model of organic matter can be expressed as [24,53]:

$$\phi_{\text{appk}} = \phi_{\text{fk}} + C_{\text{a}} \frac{ZRT}{p} \phi_{\text{ak}}, \tag{1}$$

where $\phi_{\text{appk}}$ is apparent porosity of organic matter, dimensionless; $\phi_{\text{fk}}$ is the porosity of free gas in organic matter, dimensionless; $C_{\text{a}}$ is adsorbing gas concentration on pore surface, mol/m$^3$; $Z$ is gas compression factor, dimensionless; $R$ is the general gas constant, 8.314 J/(K mol); $T$ is the reservoir temperature, K; $p$ is reservoir pressure, Pa; $\phi_{\text{ak}}$ is porosity of adsorbed gas in organic matter, dimensionless.

In the apparent porosity model, the porosity of free gas and adsorbed gas can be expressed as [24,48]:

$$\phi_{\text{fk}} = \frac{\phi_{\text{dc}} \int_{R_{\text{int\_min}}}^{R_{\text{int\_max}}} l_{\text{b}}(R_{\text{int}}) n_{\text{p}}(R_{\text{int}}) \sum\limits_{i=1}^{N} \left[ \begin{array}{c} \pi E_i \varsigma_i R_{\text{dc}}{}^2 + 4R_i \varsigma_i R_{\text{dc}}{}^2 - \\ E_i \sqrt{\vartheta_{\text{ce}} \pi \varsigma_i} \frac{R_{\text{dc}} d_{\text{m}} p}{p_{\text{L}} + p} - 4R_i (1 + \varsigma_i) \frac{R_{\text{dc}} d_{\text{m}} p}{p_{\text{L}} + p} \end{array} \right]}{\int_{R_{\text{int\_min}}}^{R_{\text{int\_max}}} l_{\text{b}}(R_{\text{int}}) n_{\text{p}}(R_{\text{int}}) \sum\limits_{i=1}^{N} \varsigma_i (\pi E_i R_{\text{dc}}{}^2 + 4R_i R_{\text{dc}}{}^2) dR_{\text{int}}}, \tag{2}$$

$$\phi_{ak} = \frac{\phi_{dc}\int_{R_{int\_min}}^{R_{int\_max}} l_b(R_{int})n_p(R_{int})\sum_{i=1}^{N}\left[E_i\sqrt{\vartheta_{ce}\pi\varsigma_i}\frac{R_{dc}d_m p}{p_L+p} + 4R_i(1+\varsigma_i)\frac{R_{dc}d_m p}{p_L+p}\right]dR_{int}}{\int_{R_{int\_min}}^{R_{int\_max}} l_b(R_{int})n_p(R_{int})\sum_{i=1}^{N}\varsigma_i(\pi E_i R_{dc}^2 + 4R_i R_{dc}^2)dR_{int}}, \tag{3}$$

where $\phi_{dc}$ is dynamic porosity of shale gas reservoirs, dimensionless [21]; $R_{int\_max}$ is the largest pore radius, m; $R_{int\_min}$ is the smallest pore radius, m; $R_{int}$ is the pore radius, m; $l_b$ is the pore length, m; $n_p$ is the pore number, dimensionless; $R_{dc}$ is the dynamic pore radius considering stress sensitivity and organic shrinkage, m; $R_i$ is the ratio of rectangular pores, dimensionless; $E_i$ is the ratio of elliptical pores, dimensionless; $\varsigma_i$ is the shape factor, dimensionless; $\vartheta_{ce}$ is the specific area, dimensionless; $d_m$ is the diameter of the gas molecular, m and $p_L$ is the Langmuir pressure, Pa.

The apparent permeability model considering the pore structure and pore size change can be expressed as [24,53]:

$$k_{appk} = \int_{R_{int\_min}}^{R_{int\_max}}\sum_{i=1}^{N}[k_{fek}(R_{int},\varsigma_i) + k_{frk}(R_{int},\varsigma_i) + k_{aek}(R_{int},\varsigma_i) + k_{ark}(R_{int},\varsigma_i)]dR_{int}, \tag{4}$$

where $k_{appk}$ is the apparent permeability of organic matter, m$^2$; $k_{fek}$ is the free gas permeability of elliptical pores in organic matter, m$^2$; $k_{frk}$ is the free gas permeability of rectangular pores in organic matter, m$^2$; $k_{aek}$ is the adsorbed gas permeability of elliptical pores in organic matter, m$^2$ and $k_{frk}$ is the adsorbed gas permeability of rectangular pores in organic matter, m$^2$, which can be expressed as:

$$k_{fek}(R_{int},\varsigma_i) = \Psi_{fem}(R_{int},\varsigma_i)F_e(R_{int},\varsigma_i)(1+\alpha Kn)\frac{\varsigma_i^2\lambda_{dc}^2}{4(\varsigma_i^2+1)}, \tag{5}$$

where $\Psi_{fem}$ is free gas permeability correction factor of porous medium with elliptical pores, dimensionless; $F_e$ is the slip factor of elliptical pores, dimensionless; $\alpha$ is correction factor for rarefaction, dimensionless; $Kn$ is the Knudsen number, dimensionless and $\lambda_{dc}$ is dynamic radius of free gas, m.

$$k_{frk}(R_{int},\varsigma_i) = \Psi_{frm}(R_{int},\varsigma_i)F_r(R_{int},\varsigma_i)(1+\alpha Kn)\frac{\lambda_{dc}^2}{3}, \tag{6}$$

where $\Psi_{frm}$ is free gas permeability correction factor of porous medium with rectangular pores, dimensionless; $F_r$ is the slip factor of rectangular pores, dimensionless.

$$k_{ark}(R_{int},\varsigma_i) = \Psi_{arm}(R_{int},\varsigma_i)D_a C_a\frac{\mu_g ZRT}{p^2}, \tag{7}$$

where $\Psi_{arm}$ is adsorbed gas permeability correction factor of porous medium with rectangular pores, dimensionless. $Da$ is the adsorption gas surface diffusion coefficient, m$^2$/s.

$$k_{aek}(R_{int},\varsigma_i) = \Psi_{aem}(R_{int},\varsigma_i)D_a C_a\frac{\mu_g ZRT}{p^2}, \tag{8}$$

where $\Psi_{aem}$ is adsorbed gas permeability correction factor of porous medium with elliptical pores, dimensionless.

### 2.2. Flow Mechanism in the Porosity of Inorganic Matter

The shale gas reservoir has low initial water saturation and ultra-low water saturation phenomenon. The gas well produces no water or produces a very small amount of water in the production process. However, in the fracturing area, due to the retention of fracturing fluid, the water saturation in the pores is higher than the irreducible water saturation, that is, there is bulk water, which leads to obvious gas–water two-phase flow in shale production. The distribution of film water and bulk water in

pores can be expressed by water saturation [53]. The research showed that water saturation has a greater impact on porosity/permeability. When the reservoir is in ultra-low water saturation, as the water saturation increases, the thickness of the water film increases, and the effective flow diameter of the gas decreases, resulting in a decrease in the gas apparent porosity/permeability. Similarly, when the reservoir is high in water saturation, an increase in water saturation leads to an increase in water phase permeability and a decrease in gas phase permeability. Considering the mechanism of micro-scale fluid migration, the static structure of porous media, the dynamic change of pore size and the distribution of water film and bulk water, the mechanism of water-gas two-phase transport in different water saturations was studied. The apparent porosity/permeability model and the gas–water relative permeability model of inorganic media with different water saturation and pore pressures are obtained.

Apparent porosity model with ultra-low water saturation can be expressed as [53]:

$$\phi_{\text{appm}} = \phi_{\text{f\_fi}} + \phi_{\text{f\_nofi}} + C_{\text{a}} \frac{ZRT}{p} \phi_{\text{a\_nofi}}, \tag{9}$$

where $\phi_{\text{appm}}$ is the apparent porosity of inorganic matter, dimensionless; $\phi_{\text{f\_fi}}$ is free gas porosity of pores filled with water film in inorganic matter, dimensionless; $\phi_{\text{f\_nofi}}$ is free gas porosity of pores without water film in inorganic matter, dimensionless and $\phi_{\text{a\_nofi}}$ is adsorbed gas porosity of pores without water film in inorganic matter, dimensionless.

Apparent permeability model with ultra-low water saturation (the original water saturation is less than the irreducible water saturation) can be expressed as [53]:

$$
\begin{aligned}
k_{\text{appm}} = {} & \int_{R_{\text{int\_min}}}^{R_{\text{fi}}-dR_{\text{int}}} \sum_{i=1}^{N} \left[ \begin{array}{l} k_{\text{fe\_nofi}}(R_{\text{int}},\varsigma_i)\Psi_{\text{fe\_nofi}}(R_{\text{int}},\varsigma_i) + k_{\text{fr\_nofi}}(R_{\text{int}},\varsigma_i)\Psi_{\text{fr\_nofi}}(R_{\text{int}},\varsigma_i) \\ +k_{\text{a}}\Psi_{\text{ae\_nofi}}(R_{\text{int}},\varsigma_i) + k_{\text{a}}\Psi_{\text{ar\_nofi}}(R_{\text{int}},\varsigma_i) \end{array} \right] dR_{\text{int}} + \\
& (1-f_{\text{Rfi}}) \sum_{i=1}^{N} \left[ \begin{array}{l} k_{\text{fe\_nofi}}(R_{\text{fi}},\varsigma_i)\Psi_{\text{fe\_nofi}}(R_{\text{fi}},\varsigma_i) + k_{\text{fr\_nofi}}(R_{\text{fi}},\varsigma_i)\Psi_{\text{fr\_nofi}}(R_{\text{fi}},\varsigma_i) \\ +k_{\text{a}}\Psi_{\text{ae\_nofi}}(R_{\text{fi}},\varsigma_i) + k_{\text{a}}\Psi_{\text{ar\_nofi}}(R_{\text{fi}},\varsigma_i) \end{array} \right] + \\
& \int_{R_{\text{fi}}+dR_{\text{int}}}^{R_{\text{int\_max}}} \sum_{i=1}^{N} [k_{\text{fe\_fi}}(R_{\text{int}},\varsigma_i)\Psi_{\text{fe\_fi}}(R_{\text{int}},\varsigma_i) + k_{\text{fr\_fi}}(R_{\text{int}},\varsigma_i)\Psi_{\text{fr\_fi}}(R_{\text{int}},\varsigma_i)] dR_{\text{int}} + \\
& f_{\text{Rfi}} \sum_{i=1}^{N} [k_{\text{fe\_fi}}(R_{\text{fi}},\varsigma_i)\Psi_{\text{fe\_fi}}(R_{\text{fi}},\varsigma_i) + k_{\text{fr\_fi}}(R_{\text{fi}},\varsigma_i)\Psi_{\text{fr\_fi}}(R_{\text{fi}},\varsigma_i)]
\end{aligned}
\tag{10}
$$

where $k_{\text{appm}}$ is the apparent permeability of inorganic matter, m$^2$; $k_{\text{fe\_nofi}}$ is the free gas permeability of elliptical pores without film water in organic matter, m$^2$; $\Psi_{\text{fe\_nofi}}$ is free gas permeability correction factor of porous medium with elliptical pores without film water, dimensionless; $k_{\text{fr\_nofi}}$ is the free gas permeability of rectangular pores without film water in organic matter, m$^2$; $\Psi_{\text{fr\_nofi}}$ is free gas permeability correction factor of porous medium with rectangular pores without film water, dimensionless; $k_{\text{a}}$ is adsorbed gas permeability, m$^2$; $\Psi_{\text{ae\_nofi}}$ is adsorbed gas permeability correction factor of porous medium with elliptical pores without film water, dimensionless; $\Psi_{\text{ar\_nofi}}$ is free gas permeability correction factor of porous medium with rectangular pores without film water, dimensionless; $k_{\text{fe\_fi}}$ is the free gas permeability of elliptical pores with film water in organic matter, m$^2$; $\Psi_{\text{fe\_fi}}$ is the free gas permeability correction factor of porous medium with elliptical pores with film water, dimensionless; $k_{\text{fr\_fi}}$ is the free gas permeability of rectangular pores with film water in organic matter, m$^2$ and $\Psi_{\text{fr\_fi}}$ is free gas permeability correction factor of porous medium with rectangular pores with film water, dimensionless.

The gas–water permeability model with high water saturation (the original water saturation is larger than the irreducible water saturation) can be expressed as [53]:

$$k_{\text{gas}} = \int_{R_{\text{bw\_lim}}}^{R_{\text{int\_max}}} \sum_{i=1}^{N} \left[ k_{\text{gas\_part\_e}}(R_{\text{int}},\varsigma_i) + k_{\text{gas\_part\_r}}(R_{\text{int}},\varsigma_i) \right] dR_{\text{int}}, \tag{11}$$

$$k_{\text{water}} = \int_{R_{\text{int\_min}}}^{R_{\text{bw\_lim}}} \sum_{i=1}^{N} [k_{\text{fule}}(R_{\text{int}}, \varsigma_i) + k_{\text{fulr}}(R_{\text{int}}, \varsigma_i)] dR_{\text{int}} + \\ \int_{R_{\text{bw\_lim}}}^{R_{\text{int\_max}}} \sum_{i=1}^{N} \left[ k_{\text{water\_part\_e}}(R_{\text{int}}, \varsigma_{\text{bw}}) + k_{\text{water\_part\_r}}(R_{\text{int}}, \varsigma_{\text{bw}}) \right] dR_{\text{int}} \tag{12}$$

where $k_{\text{gas}}$ is the gas permeability of inorganic matter with high water saturation, m$^2$; $k_{\text{gas\_part\_e}}$ is the gas permeability of elliptical pores of inorganic matter partly filled with bulk water, m$^2$; $k_{\text{gas\_part\_r}}$ is the gas permeability of rectangular pores of inorganic matter partly filled with bulk water, m$^2$; $k_{\text{water}}$ is the water permeability of inorganic matter with high water saturation, m$^2$; $k_{\text{water\_part\_e}}$ is the water permeability of elliptical pores of inorganic matter partly filled with bulk water, m$^2$; $k_{\text{water\_part\_r}}$ is the water permeability of rectangular pores of inorganic matter partly filled with bulk water, m$^2$; $k_{\text{fule}}$ is the water permeability of elliptical pores of inorganic matter fully filled with bulk water, m$^2$ and $k_{\text{fulr}}$ is the water permeability of rectangular pores of inorganic matter partly fully with bulk water, m$^2$.

### 2.3. Flow Mechanism in Induced Fractures

The distribution of induced fractures after reservoir fracturing is complex. It is difficult to accurately describe because it is the result of a comprehensive influence of rock mechanics parameters, fracturing parameters and natural fracture distribution of reservoirs. The distribution of induced fractures in ESRV was quantitatively characterized by induced fractures spacing/aperture fractal dimension, and then the porosity/permeability characterization method of fracture system in the ESRV region was established [45].

$$\phi_{\text{f}} = \phi_i \frac{s_{\text{m}}(x)}{s_{\text{m}}(x) + s_{\text{f}}(x)} = \phi_{\text{w}} \left( \frac{x}{x_{\text{w}}} \right)^{d_{\text{fs}} + d_{\text{fa}} - 4}, \tag{13}$$

$$k_{\text{f}}(x) = k_{\text{w}} \left( \frac{x}{x_{\text{w}}} \right)^{3d_{\text{fa}} - 6} \left( \frac{x}{x_{\text{w}}} \right)^{d_{\text{fs}} - 2} \left( \frac{x}{x_{\text{w}}} \right)^{-\theta}, \tag{14}$$

where $\phi_{\text{f}}$ is the porosity of the fracture system, dimensionless; $\phi_i$ is the porosity of induced fractures, dimensionless; $s_m$ is the induced fractures aperture, m; $s_f$ is the induced fractures apscing, m; $\phi_{\text{w}}$ is the porosity of the fracture system near the hydraulic fracture, dimensionless; $x_{\text{w}}$ is the reference point, m; $d_{\text{fs}}$ is the induced fractures spacing fractal dimension, dimensionless; $d_{\text{fa}}$ is the induced fractures aperture fractal dimension, dimensionless; $k_{\text{f}}$ is the permeability of the fracture system, m$^2$; $k_{\text{w}}$ is the permeability of the fracture system near hydraulic fracture, m$^2$; $\theta$ is the fractal index, dimensionless.

Considering that the scale of induced fractures in the shale gas reservoir is relatively large, and the influence of water saturation on water and gas flow can be characterized by the relative permeability curve of the conventional gas reservoir. Therefore, the effect of water saturation on porosity/permeability of induced fractures in shale gas reservoir is not carried out in this paper.

## 3. Mathematical Model Establishment

### 3.1. Physical Model of Multi-Fractured Horizontal Wells

As described above, considering the influence of fracturing water retention in the ESRV region, the porous medium of the inorganic system in ESRV region has high water saturation. The control area of a single hydraulic fracture of multi-fractured horizontal wells was analyzed in this part, and a multi-porosity media model considering gas and water flow for the multi-fractured horizontal wells was proposed, as shown in Figure 1.

Region 1 represents the hydraulic fracture, which is a single-porosity medium with the gas–water flow. Region 2 represents the ESRV, filled with triple-porosity media, including induced fractures, inorganic matter and organic matter. In the ESRV, the induced fractures and inorganic matter are assumed as filled with bulk water and gas, and in the pores of organic matter, only gas is storage. Region 3 is USRV, filled with dual-porosity media, including inorganic matter and organic matter.

As analyzed above, only original water is distributed in the pores of inorganic matter, and the original water saturation is generally lower than irreducible water saturation. Therefore, only gas flow is considered in the pores of inorganic matter, and the water is considered as irreducible water. Same as the pores of organic matter in ESRV, only gas exists in the pores of organic matter in USRV.

The basic assumptions of the model are as follows. The shale gas reservoir is a closed rectangular reservoir with a multi-fractured horizontal well. The hydraulic fractures are evenly distributed, and the reservoir in the middle of adjacent hydraulic fractures is completely stimulated. The control area of a single hydraulic fracture of multi-fractured horizontal wells was analyzed, and the model could be extended to the entire multi-fractured horizontal well by modifying the boundary conditions. In the USRV (Region 3), there were dual-porosity media, including organic matter with oil-wet pores and inorganic matter with water-wet pore. In the pores of inorganic matter, the pore surface covered the water film, and the film water did not flow during development. In the ESRV, there were triple-porosity media, including organic matter, inorganic matter and induced fractures. The properties of the organic matter in ESRV were the same as that in USRV. In the pores of inorganic matter in ESRV, there were film water and bulk water resulting from hydraulic fracturing practice, and the bulk water will flow during the production process. The induced fractures in the ESRV were heterogeneously distributed, and the distribution of induced fractures were assumed that the induced fractures spacing and aperture were consistent with the fractal distribution. The hydraulic fracture penetrated the entire shale gas reservoir, and the fracture was limited conductivity. The two-phase flow of water and gas was considered in the hydraulic fracture. The water saturation in the inorganic pores and induced fractures in the ESRV and hydraulic fracture was the same. During the production of shale gas reservoirs, the reservoirs were kept isothermal and did not take into account the effects of gravity and other factors. The proposed model was an ideal two-dimension (2D) model, which only considered the complex physical distribution in a 2D domain, and further work could be done to propose a three-dimension (3D) model to describe the actual reservoir more accurately. However, in the 3D model, some more complex phenomenon should be fully discussed, such as the non-rectangular hydraulic fractures, the distribution of induced fractures in the vertical direction of the reservoir.

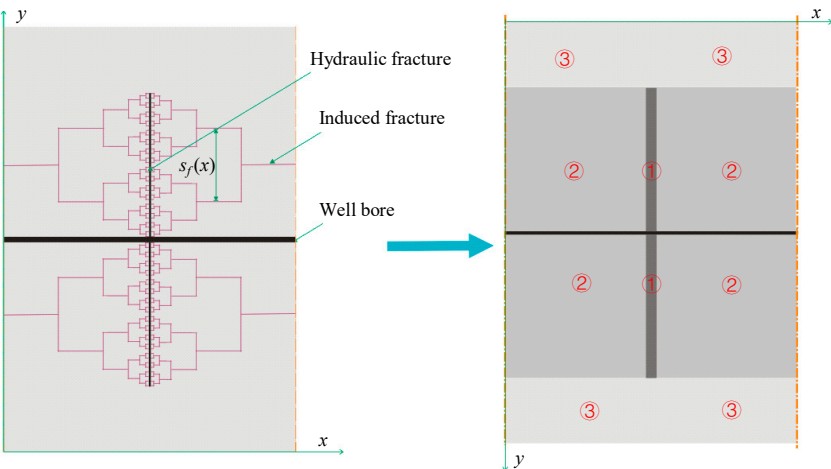

**Figure 1.** The physical model of multi-phase flow in multi-porosity media of a multi-fractured horizontal well.

*3.2. Mathematical Model in Multi-Porosity Media of Shale Gas Reservoirs*

3.2.1. Single-Phase Flow in Dual-Porosity Media of USRV

It is assumed that in the dual-porosity media of organic matter and inorganic matter, the mass transfer from organic matter to inorganic matter is a pseudo-steady flow. As discussed above, only gas flow was considered in this region. Based on the apparent porosity/permeability method established

in the previous work, the flow characteristics of various migration mechanisms in shale gas reservoirs could be described. The flow equations can be expressed as:

$$
\begin{cases}
\dfrac{\partial\left(\rho_g\phi_{3k}\right)}{\partial t} = -\rho_g\sigma_k\dfrac{k_k}{\mu_g}\left(p_{3k}-p_{3m}\right) \\
\dfrac{\partial\left(\rho_g\phi_{3m}\right)}{\partial t} = \dfrac{\partial}{\partial x}\left(\rho_g\dfrac{k_k}{\mu_g}\dfrac{\partial p_{3m}}{\partial x}\right) + \dfrac{\partial}{\partial y}\left(\rho_g\dfrac{k_k}{\mu_g}\dfrac{\partial p_{3m}}{\partial y}\right) + \rho_g\sigma_k\dfrac{k_k}{\mu_g}\left(p_{3k}-p_{3m}\right)
\end{cases}, \tag{15}
$$

where subscripts 2 and 3 represent the properties of the ESRV region and the USRV region, respectively. The subscripts k and m represent the properties of organic matter and inorganic smatter, respectively. The porosity/permeability of organic matter and inorganic system can be calculated from Equation (1), Equation (4), Equation (9) and Equation (10), respectively.

### 3.2.2. Two Phase Flow in Triple-Porosity Media of ESRV

Induced fractures, inorganic matter and organic matter are developed in the ESRV region, the distribution and flow pattern of the water–gas are complex. According to the previous analysis, we made the following assumptions. The gas transfer from organic matter to inorganic matter was a pseudo-steady flow. The flow of gas in the inorganic porous medium could be described by the apparent permeability/porosity model, at the same time, the pores had the gas influx from the organic matter with the pseudo-steady flow and gas outflow to induced fractures, which could also be described by the pseudo-steady flow. There was bulk water and film water on the pore surface in the inorganic matter. The film water did not flow during development, while the bulk water flows in the pores and then transfer to induced fractures with a pseudo-steady flow. The water–gas two-phase flows in the induced fractures were a two-dimensional flow, and had the mass transfer of water–gas from inorganic matter.

The water–gas flow equations in organic matter, inorganic matter and induced fractures can be characterized by the apparent permeability/porosity model from the above section. Therefore, the flow in the triple-porosity media of ESRV can be described by the following equations.

$$
\begin{cases}
\dfrac{\partial\left(\rho_g\phi_{2k}\right)}{\partial t} = -\rho_g\sigma_k\dfrac{k_k}{\mu_g}\left(p_{2k}-p_{2m}\right) \\
\dfrac{\partial\left(\rho_w\phi_{2m}S_w\right)}{\partial t} = -\rho_w\sigma_k\dfrac{k_m k_{rw}}{\mu_w}\left(p_{2m}-p_{2f}\right) \\
\dfrac{\partial\left(\rho_g\phi_{2m}S_g\right)}{\partial t} = -\rho_g\sigma_k\dfrac{k_m k_{rg}}{\mu_g}\left(p_{2m}-p_{2f}\right) + \rho_g\sigma_k\dfrac{k_k}{\mu_g}\left(p_{2k}-p_{2m}\right) \\
\dfrac{\partial\left[\rho_w\phi_f(x)S_w\right]}{\partial t} = \dfrac{\partial}{\partial x}\left[\rho_w\dfrac{k_f(x)k_{rw}}{\mu_w}\dfrac{\partial p_f}{\partial x}\right] + \dfrac{\partial}{\partial y}\left[\rho_w\dfrac{k_f(x)k_{rw}}{\mu_w}\dfrac{\partial p_f}{\partial y}\right] + \rho_w\sigma_k\dfrac{k_m k_{rw}}{\mu_w}\left(p_{2m}-p_{2f}\right) \\
\dfrac{\partial\left[\rho_g\phi_f(x)S_g\right]}{\partial t} = \dfrac{\partial}{\partial x}\left[\rho_g\dfrac{k_f(x)k_{rg}}{\mu_g}\dfrac{\partial p_f}{\partial x}\right] + \dfrac{\partial}{\partial y}\left[\rho_g\dfrac{k_f(x)k_{rg}}{\mu_g}\dfrac{\partial p_f}{\partial y}\right] + \rho_g\sigma_k\dfrac{k_m k_{rg}}{\mu_g}\left(p_{2m}-p_{2f}\right)
\end{cases}, \tag{16}
$$

where $k_{rw}$ represents the relative permeability of the water phase and $k_{rg}$ represents the relative permeability of the gas phase.

### 3.2.3. Two Phase Flow in Single-Porosity Media of Hydraulic Fracture

The hydraulic fracture is a single-porosity medium, and there is gas–water two-phase flow. The flow equation is:

$$
\begin{cases}
\dfrac{\partial\left[\rho_w\phi_F(x)S_w\right]}{\partial t} = \dfrac{\partial}{\partial x}\left(\rho_w\dfrac{k_F k_{rw}}{\mu_w}\dfrac{\partial p_F}{\partial x}\right) + \dfrac{\partial}{\partial y}\left(\rho_w\dfrac{k_F k_{rw}}{\mu_w}\dfrac{\partial p_F}{\partial y}\right) \\
\dfrac{\partial\left[\rho_g\phi_F(x)S_g\right]}{\partial t} = \dfrac{\partial}{\partial x}\left(\rho_g\dfrac{k_F k_{rg}}{\mu_g}\dfrac{\partial p_F}{\partial x}\right) + \dfrac{\partial}{\partial y}\left(\rho_g\dfrac{k_F k_{rg}}{\mu_g}\dfrac{\partial p_F}{\partial y}\right)
\end{cases}, \tag{17}
$$

where the subscript F represents the properties of hydraulic fracture.

3.2.4. Model Solution and Validation

The finite difference method was used to solve the whole model. According to the previous analysis, the pressure and parameters of the induced fractures in the area around the hydraulic fracture were greatly heterogeneous, therefore, the index mesh was used in the meshing. At the same time, when designing the time step, consider the dynamic change of the bottom hole in the initial stage of production, and select the exponential change mode for the time step design. Compared with the existing commercial software, the multi-media water–gas two-phase flow model established in this paper could consider the micro-scale coupling transport mechanism in organic and inorganic media, and could accurately consider the porosity and permeability of different media with pressure variation. The correctness of the model requires verification.

Based on numerical simulation software, we could establish a multi-media water–gas two-phase flow model with media properties that did not change with pressure. Regardless of the change of apparent porosity/permeability of organic matter and inorganic matter with pressure, the flow behavior was calculated by the proposed model. The results were compared with the numerical simulation results, and the results are shown in Figure 2. The reservoir parameters used in the model validation are listed in Table 1, and the property parameter for apparent porosity/permeability model could be obtained from Sheng's dissertation [53]. It can be seen from the figure that when the proposed model did not consider the organic and inorganic properties with pressure variation, the model calculation results were in good agreement with the numerical simulation results. When considering the change of properties of organic matter and inorganic matter with pressure, the gas production was relatively small after 100 days of production compared with the numerical simulation calculation. The main reason was that when the reservoir pressure dropped, according to the previous analysis, the organic/inorganic permeability was reduced and thus its production was also reduced. It can be seen from the model comparison that the calculation results of this model were reliable, and properties of organic matter and inorganic matter, which were greatly affected by pore pressure, had a certain influence on the production, and should be considered in production analysis.

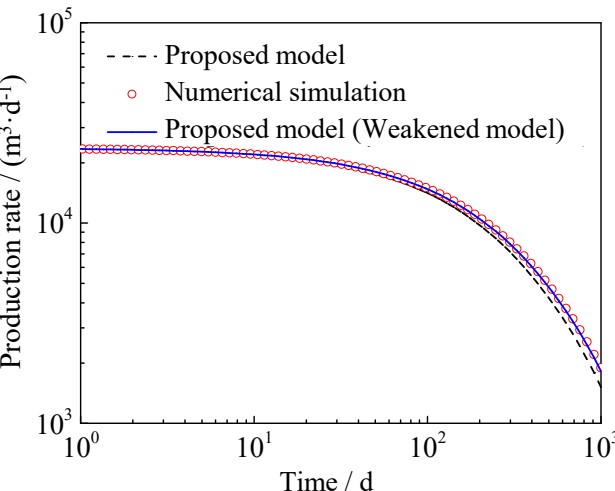

**Figure 2.** Comparison of results from numerical simulation and the proposed model.

**Table 1.** Reservoir parameters for model validation and flow behavior analysis.

| Parameters | Value | Parameters | Value |
|---|---|---|---|
| Half-length of hydraulic fracture/m | 83.82 | Spacing of induced fractures near hydraulic fracture/m | 0.17 |
| Half-aperture of hydraulic fracture/m | 0.00152 | Half-width of reservoirs/m | 167.64 |

**Table 1.** *Cont.*

| Parameters | Value | Parameters | Value |
|---|---|---|---|
| Half-spacing between hydraulic fractures/m | 27.52 | Reservoir thickness/m | 91.44 |
| Permeability of hydraulic fracture/mD | 100 | Gas viscosity/cp | 0.0184 |
| Porosity of hydraulic fracture | 0.38 | Reservoir temperature/K | 314.26 |
| Porosity of induced fractures near hydraulic fracture | 0.8 | Porosity of organic matter | 0.2 |
| Aperture of induced fractures near hydraulic fracture/m | 0.000152 | Total organic content (TOC) | 0.2 |
| Porosity of inorganic matter | 0.05 | Initial water saturation in inorganic matter | 0.1152 |
| Irreducible water saturation | 0.1952 | Water saturation in effectively stimulated reservoir volume (ESRV) | 0.5 |

## 4. Analysis and Discussion

### 4.1. Flow Behavior Analysis

Based on the proposed model, the water and gas production at the bottom of the well was calculated. The results are shown in Figure 3. As can be seen from the figure, based on the data in this paper, the initial gas production in the single hydraulic fracture control area was 23,000 square meters, and the initial water production was about 3.25 square meters. With the gradual increase of production time, the gas production rate decreased rapidly relative to the water production rate, and the gas production rate gradually flattened after 400 days of production. It should be noted that after 1000 days, the decrease in water production was still relatively fast, and the reservoir had not yet reached a steady state.

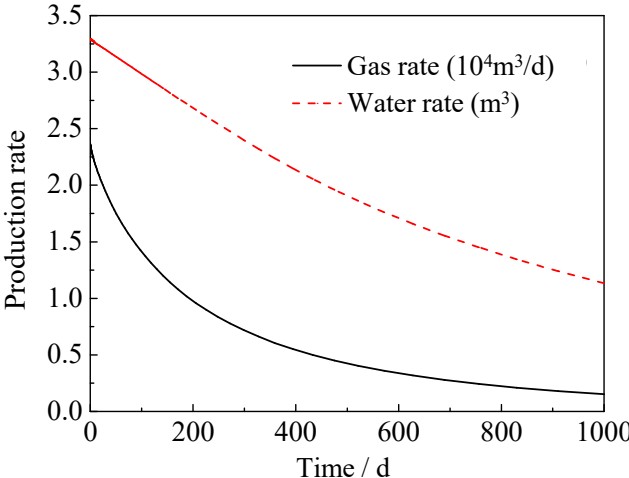

**Figure 3.** The flow rate of multi-fractured horizontal wells in shale gas reservoirs.

Figure 4 shows the distribution of gas saturation in inorganic matter at 1 day of production. It can be seen from the figure that the USRV region had the same gas saturation as the initial gas saturation because it did not consider the water phase migration, and did not change with production. The closer the region to the hydraulic fracture, the lower the gas saturation and the higher the relative water saturation. The variation of gas saturation in inorganic matter in the ESRV region with production time was analyzed. The results are shown in Figure 5. It can be seen from the figure that as the production time increased, the gas saturation in the inorganic matter of the ESRV region gradually decreased. After 100 days of production, the gas saturation of the ESRV region was close to about 0.2, and with

the further increase of time, the change in gas saturation gradually slowed down, and the rate of gas production declined gradually.

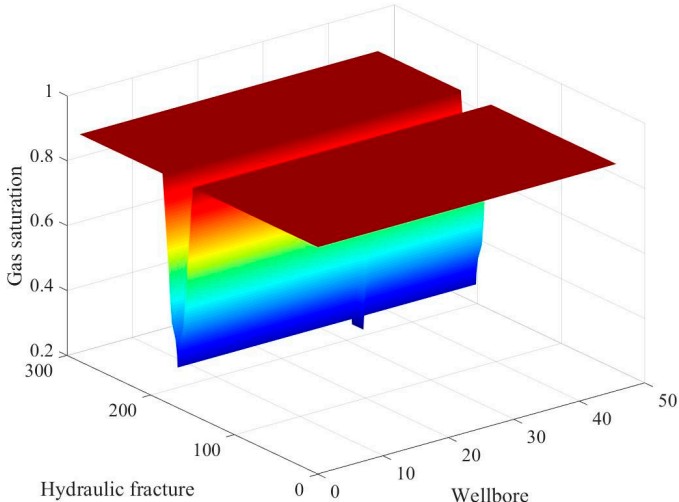

**Figure 4.** The distribution of gas saturation after 1 day.

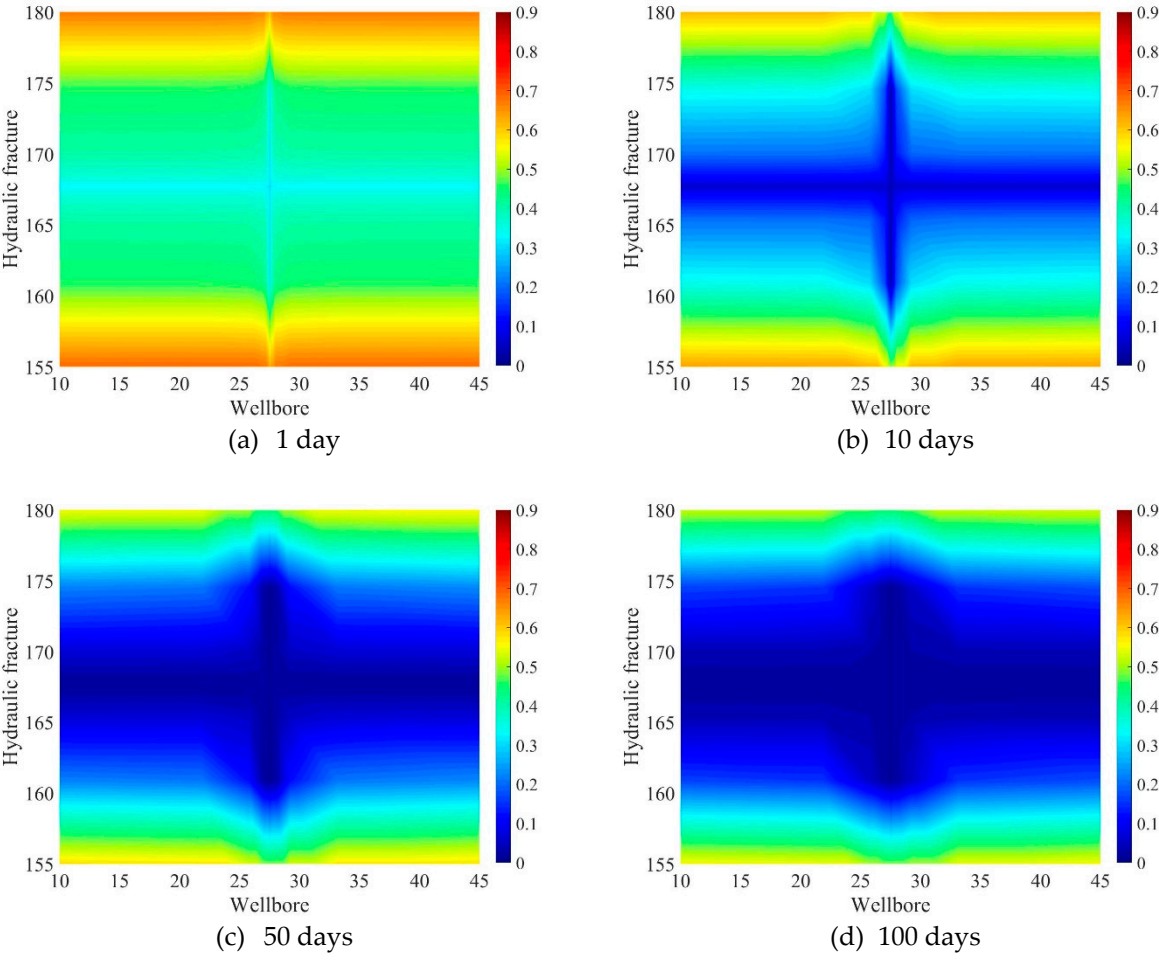

**Figure 5.** The distribution of gas saturation in inorganic matrix in effectively stimulated reservoir volume (ESRV).

## 4.2. Production Analysis of Shale Gas Reservoirs

Based on the proposed model, the production rate of multi-fractured horizontal wells was analyzed, and the typical parameters of organic matter and inorganic matter were selected to analyze the factors affecting production.

### 4.2.1. Influence of Total Organic Content (TOC)

Assuming that the total organic content (TOC) was 0.1, 0.2 and 0.3 respectively, the gas production of single hydraulic fracture was calculated based on the model, and the influence of TOC on gas production was analyzed. The results are shown in Figure 6. It can be seen from the figure that the TOC mainly affected gas production after 40 days. With the increase of production time, the influence of TOC gradually increased, and the higher the TOC, the greater the gas production at the later stage. The main reason was that the inherent porosity of the organic matter was relatively large compared with the inorganic matter, and the apparent permeability of the organic matter and inorganic matter was similar. The increase of the TOC increased the original gas content of the reservoir, which in turn increased the gas production in the later stage of production.

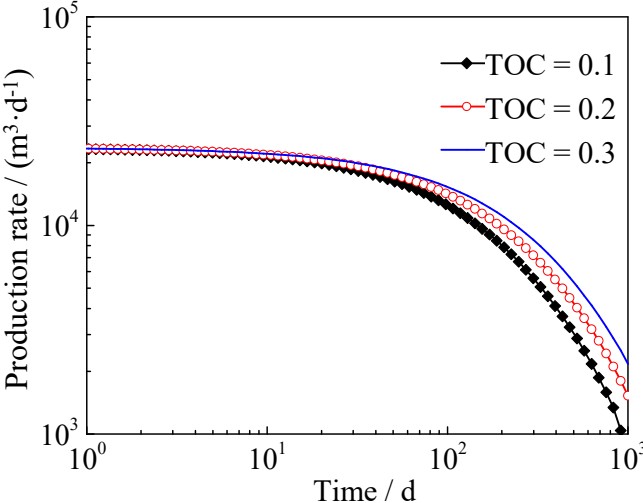

**Figure 6.** Influence of the total organic content (TOC) on the gas flow rate in shale gas reservoirs.

### 4.2.2. Influence of the Inherent Porosity of Organic Matter

It was assumed that the inherent porosity of organic matter was 0.2, 0.4 and 0.6 respectively. Based on the proposed model, the gas production of a single hydraulic fracture was calculated, and the influence of organic porosity on gas production was analyzed. The results are shown in Figure 7. It can be seen from the figure that the influence of organic porosity on the production also occurred after 40 days of production, and with the increase of production time, the influence gradually increased. The larger the inherent porosity of organic matter, the higher the gas production of shale gas reservoirs. As the inherent porosity increases, the increase in gas production was getting smaller and smaller. The main reason was that the increase of the inherent porosity of the organic matter means the increase of the original gas reservoir, which in turn increased the gas production of the production well.

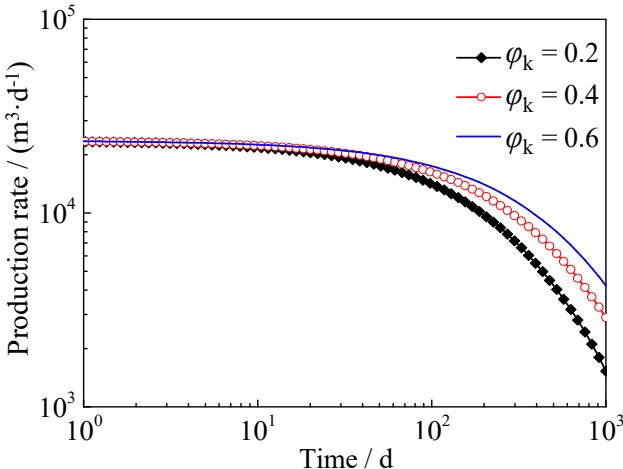

**Figure 7.** Influence of organic porosity on gas flow.

### 4.2.3. Influence of Pore Size Change

Based on the analysis of organic matter properties, it can be seen that the pore size during depressurization was affected by the stress sensitivity and the organic matter shrinkage effect. Based on the model of this paper, we calculated the production with three cases: The stress sensitivity and organic matter shrinkage were both considered (Case 1), did not consider stress sensitivity and only considered organic matter shrinkage (Case 2) and did not consider shrinkage and only considered stress sensitivity (Case 3). The results are shown in Figure 8. As can be seen from the figure, when stress sensitivity was not considered, the gas production would increase after 400 days. The main reason was that when the influence of stress sensitivity was not considered, the pore size shrinkage caused by the stress was neglected, so gas production in the early period increased. As the production time increased, the pore pressure gradually decreased, the influence of stress sensitivity gradually increased. When the effect of organic matter shrinkage was not considered, the gas production would gradually decrease after 500 days. The main reason was that the organic matter shrinkage increased the pore size. When the influence of organic matter shrinkage was neglected, the shrinkage of organic matter was neglected, and the process of increasing pore size was not considered, resulting in a small calculation of gas production. Similarly, as production time increased, the effect of organic matter shrinkage increased.

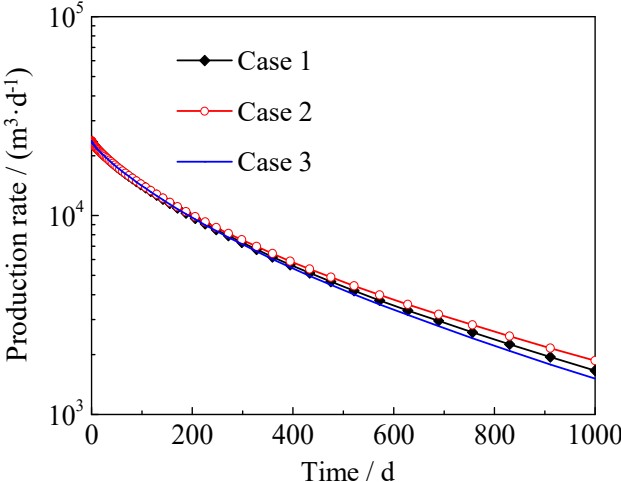

**Figure 8.** Influence of radius-changing during decompression production on gas flow rate.

#### 4.2.4. Influence of Water Saturation with Ultra-Low Water Saturation

The shale gas reservoir had a small amount of water in the pores during the accumulation process, and was present in the pore surface as a water film. The presence of water film will cause the gas flow channel to decrease, and at the same time, the gas will not be adsorbed on the pore surface, which will have a greater impact on gas migration. The previous studies show that the film water saturation is about 0.2, and the gas production is calculated at different water saturations, assuming the water saturation is 0.01, 0.1 and 0.1952, respectively. The results are shown in Figure 9. It illustrates that the film water in ultra-low water saturation mainly affected gas production after 100 days. The greater the water saturation, the smaller the gas production, and the influence of film water was gradually increasing. In general, the presence of water film will have a certain adverse effect on the production of shale gas reservoirs, but it can be seen that when the water saturation was less than 0.1, the impact was small.

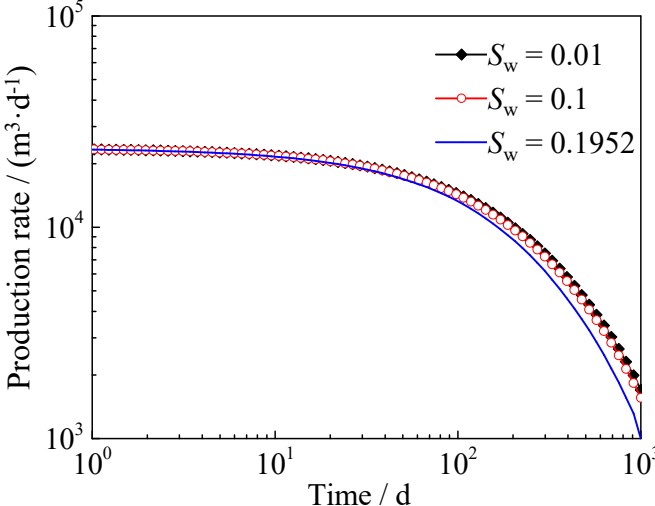

**Figure 9.** Influence of water saturation on gas flow rate in shale gas reservoirs with ultra-low water saturation.

#### 4.2.5. Influence of Water Saturation in ESRV

In shale gas reservoirs, the water saturation will increase due to the retention of the fracturing fluid in the ESRV, and the water film will be bound to form bulk water. Assuming that the water saturation of the ESRV region was 0.4, 0.5 and 0.6, respectively, the influence of bulk water on the gas production of the shale gas reservoir was analyzed. The results are shown in Figure 10. It can be seen from the figure that the bulk water in the ESRV region had a great influence on the gas production of the shale gas reservoir in the whole stage. With the increase of water saturation, the gas production in the shale gas reservoir decreased overall. When the bulk water saturation reached a certain level (the water saturation limit of this paper was 0.6), the gas production would decrease sharply in the later stage of production. The main reason was that when the water saturation was large, with the increase of production time, the water saturation in the near-well zone gradually increased, and the gas saturation gradually decreased. When it increased to a certain extent, the gas flow channel decreased sharply, resulting in a sharp decrease.

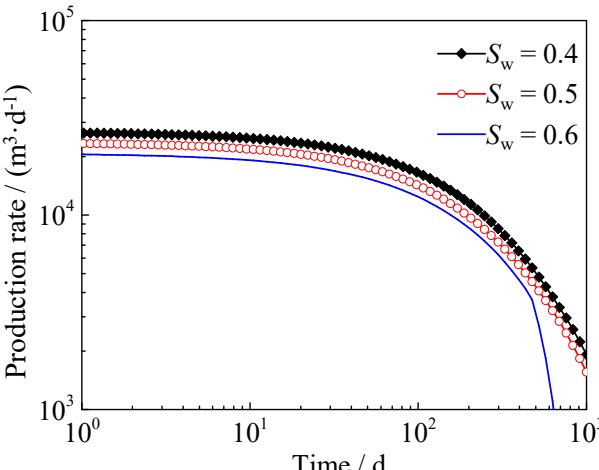

**Figure 10.** Influence of water saturation in stimulated reservoir volume (SRV) on the gas flow rate in a shale gas reservoir with high water saturation.

## 5. Conclusions

In this paper, a multi-media gas–water two-phase numerical model of multi-fractured horizontal wells in shale gas reservoirs was established by combining with the apparent porosity/permeability model of organic matter, inorganic matter, induced fractures and hydraulic fractures. The water–gas two-phase flow behavior of multi-fractured horizontal wells was obtained and the effects of reservoir parameters on gas production were discussed. The main conclusions were as follows:

(1) The flow characteristics in fractured shale reservoirs were complex. In the USRV region, the organic matter only existed in the gas phase, and pores in the inorganic matter had water film and gas. In the ESRV region, organic matter pores had oil wetness characteristics, only considering the gas phase flow. The inorganic matrix pores had film water and bulk water, and there was a two-phase flow of water and gas. In the hydraulic fracture, the two-phase flow of water and gas was considered.

(2) Based on the proposed model, the two-phase flow behavior of multi-fractured horizontal wells was studied. The results show that the USRV region had the same gas saturation as the initial gas saturation because it did not consider the water flow. As the production time increased, the gas saturation in the inorganic matter of the ESRV region gradually decreased. After 400 days of production, the gas saturation of the ESRV region was close to about 0.2.

(3) Based on the proposed model, the gas production of multi-fractured horizontal wells was analyzed. The results showed that the TOC and inherent porosity of organic matter had an effect on the production of gas after 40 days, and the influence increased with the increase of production time.

(4) When stress sensitivity was not considered, it would be caused the gas production increase after 400 days. When the organic matter shrinkage was not taken into account, the gas production would gradually decrease after 500 days. The water film mainly affected gas production after 100 days of production. The bulk water in ESRV region had a greater impact on the gas production of shale gas reservoirs throughout the whole period.

**Author Contributions:** Methodology, Writing—review & editing, L.L.; Supervision, Funding Acquisition, Y.S.; Investigation, Data Curation, G.S.

**Funding:** The work is supported by China Postdoctoral Science Foundation (2018M630813, 2019T120616), the National Natural Science Foundation of China (51904324, 51974348), the China Major National Science and Technology Project (2017ZX05009, 2017ZX05072), the Fundamental Research Funds for the Central Universities (18CX02170A), the Postdoctoral Applied Research Project Foundation of Qingdao city (BY201802003), and the Funding for Scientific Research of China University of Petroleum East China (YJ20170013).

**Conflicts of Interest:** The authors declare no conflicts of interest.

## Nomenclature

| | | |
|---|---|---|
| ESRV | effectively stimulated reservoir volume | |
| FDE | fractal diffusion equation | |
| USRV | unstimulated reservoir volume | |
| TOC | total organic content | |
| $\alpha$ | correction factor for rarefaction | dimensionless |
| $C_a$ | adsorbing gas concentration on pore surface | mol/m$^3$ |
| $Da$ | adsorption gas surface diffusion coefficient | m$^2$/s |
| $d_m$ | diameter of gas molecular | m |
| $d_{fa}$ | induced fractures aperture fractal dimension | dimensionless |
| $d_{fs}$ | induced fractures spacing fractal dimension | dimensionless |
| $E_i$ | he ratio of elliptical pores | dimensionless |
| $F_e$ | slip factor of elliptical pores | dimensionless |
| $F_r$ | the slip factor of rectangular pores | dimensionless |
| $k_a$ | adsorbed gas permeability | m$^2$ |
| $k_{appk}$ | apparent permeability of organic matter | m$^2$ |
| $k_{appm}$ | apparent permeability of inorganic matter | m$^2$ |
| $k_{aek}$ | adsorbed gas permeability of elliptical pores in organic matter | m$^2$ |
| $k_f$ | permeability of fracture system | m$^2$ |
| $k_{fe\_fi}$ | free gas permeability of elliptical pores with film water in organic matter | m$^2$ |
| $k_{fek}$ | free gas permeability of elliptical pores in organic matter | m$^2$ |
| $k_{fe\_nofi}$ | free gas permeability of elliptical pores without film water in organic matter | m$^2$ |
| $k_{fr\_nofi}$ | free gas permeability of rectangular pores without film water in organic matter | m$^2$ |
| $k_{frk}$ | free gas permeability of rectangular pores in organic matter | m$^2$ |
| $k_{frk}$ | adsorbed gas permeability of rectangular pores in organic matter | m$^2$ |
| $k_{gas}$ | gas permeability of inorganic matter with high water saturation | m$^2$ |
| $k_{gas\_part\_e}$ | gas permeability of elliptical pores of inorganic matter partly filled with bulk water | m$^2$ |
| $k_{gas\_part\_r}$ | gas permeability of rectangular pores of inorganic matter partly filled with bulk water | m$^2$ |
| $k_{water}$ | water permeability of inorganic matter with high water saturation | m$^2$ |
| $k_{water\_part\_e}$ | water permeability of elliptical pores of inorganic matter partly filled with bulk water | m$^2$ |
| $k_{water\_part\_r}$ | water permeability of rectangular pores of inorganic matter partly filled with bulk water | m$^2$ |
| $k_{fule}$ | water permeability of elliptical pores of inorganic matter fully filled with bulk water | m$^2$ |
| $k_{fulr}$ | water permeability of rectangular pores of inorganic matter partly fully with bulk water | m$^2$ |
| $k_w$ | permeability of fracture system near hydraulic fracture | m$^2$ |
| $k_{rw}$ | the relative permeability of the water phase | dimensionless |
| $k_{rg}$ | the relative permeability of the gas phase | dimensionless |
| $Kn$ | Knudsen number | dimensionless |
| $l_b$ | pore length | m |
| $n_p$ | pore number | dimensionless |
| $p$ | reservoir pressure | Pa |
| $p_L$ | Langmuir pressure | Pa |
| R | the general gas constant | 8.314 J/(K mol) |
| $R_{int\_max}$ | the largest pore radius | m |
| $R_{int\_min}$ | the smallest pore radius | m |
| $R_{int}$ | pore radius | m |
| $R_{dc}$ | dynamic pore radius considering stress sensitivity and organic shrinkage | m |
| $R_i$ | the ratio of rectangular pores | dimensionless |
| $s_m$ | induced fractures aperture | m |
| $s_f$ | induced fractures apscing | m |
| $T$ | reservoir temperature | K |
| $x_w$ | the reference point | m |
| $Z$ | gas compression factor | dimensionless |
| $\phi_{appk}$ | apparent porosity of organic matter | dimensionless |
| $\phi_{a\_nofi}$ | adsorbed gas porosity of pores without water film in inorganic matter | dimensionless |
| $\phi_{fk}$ | porosity of free gas in organic matter | dimensionless |
| $\phi_{ak}$ | porosity of adsorbed gas in organic matter | dimensionless |
| $\phi_{dc}$ | dynamic porosity of shale gas reservoirs | dimensionless |
| $\phi_{appm}$ | apparent porosity of inorganic matter | dimensionless |
| $\phi_f$ | porosity of fracture system | dimensionless |
| $\phi_{f\_fi}$ | free gas porosity of pores filled with water film in inorganic matter | dimensionless |
| $\phi_{f\_nofi}$ | free gas porosity of pores without water film in inorganic matter | dimensionless |
| $\phi_i$ | porosity of induced fractures | dimensionless |
| $\phi_w$ | porosity of fracture system near hydraulic fracture | dimensionless |
| $\Psi_{fem}$ | free gas permeability correction factor of porous medium with elliptical pores | dimensionless |

| | | |
|---|---|---|
| $\Psi_{frm}$ | free gas permeability correction factor of porous medium with rectangular pores | dimensionless |
| $\Psi_{arm}$ | adsorbed gas permeability correction factor of porous medium with rectangular pores | dimensionless |
| $\Psi_{aem}$ | adsorbed gas permeability correction factor of porous medium with elliptical pores | dimensionless |
| $\Psi_{fe\_nofi}$ | free gas permeability correction factor of porous medium with elliptical pores without film water | dimensionless |
| $\Psi_{fr\_nofi}$ | free gas permeability correction factor of porous medium with rectangular pores without film water | dimensionless |
| $\Psi_{ae\_nofi}$ | adsorbed gas permeability correction factor of porous medium with elliptical pores without film water | dimensionless |
| $\Psi_{ar\_nofi}$ | free gas permeability correction factor of porous medium with rectangular pores without film water | dimensionless |
| $\Psi_{fe\_fi}$ | free gas permeability correction factor of porous medium with elliptical pores with film water | dimensionless |
| $k_{fr\_fi}$ | free gas permeability of rectangular pores with film water in organic matter | $m^2$ |
| $\Psi_{fr\_fi}$ | free gas permeability correction factor of porous medium with rectangular pores with film water | dimensionless |
| $\lambda_{dc}$ | dynamic radius of free gas | m |
| $\theta$ | fractal index | dimensionless |
| $\varsigma_i$ | shape factor | dimensionless |
| $\vartheta_{ce}$ | specific area | dimensionless |

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
