# Peer review of "Water-Gas Two-Phase Flow Behavior of Multi-Fractured Horizontal Wells in Shale Gas Reservoirs"

_processes, doi:10.3390/pr7100664_

Round 1

Reviewer 1 Report

In the paper, a multi-media water-gas two-phase flow model considering water film in organic pores, inorganic pores, and bulk water in ESRV was proposed. The finite difference method is used and the water-gas two-phase flow behavior of multi-fractured horizontal wells is obtained. The research can describe the water-gas flow behavior in different scale media of shale gas reservoirs. The work is important to the energy industry as shale oil and gas recovery represents much of the recovery in the world currently and perhaps more in the future. I recommend the work to be published after the following points can be modified.

In this paper, flow mechanisms in organic matter, inorganic matter, and induced fractures are simulated by apparent porosity/permeability and fractal porosity/permeability respectively. The research progress about fractal porosity/permeability for induced fracture has been fully discussed in the introduction. However, the research progress about apparent porosity/ permeability of organic and inorganic matter are not discussed in the introduction. The research progress and the application of apparent porosity/permeability on flow simulation should be introduced in the paper. Section 2.2. The water film and bulk water in organic pores are considered, and porosity/permeability with ultra-low water saturation and high water saturation are proposed. So, how to determine whether the reservoir is ultra-low water saturation or high water saturation? Does water saturation have an effect on porosity/permeability? There is no water saturation in Eq.9-12, how to reflect the effect of water saturation on permeability? Section 2.3. Considering the hydraulic fracturing practice, the induced fractures have bulk water. The effect of water saturation on porosity/permeability of induced fracture should be given to simulate water-gas flow in the fractures. Section 3.2. Are the water-gas relative permeability in Eq.16 for organic pores and induced fracture the same? And are the water-gas relative permeability in Eq.16 and Eq.17 the same? If not, please use different symbols to represent and introduce the calculation methods.

Reviewer 2 Report

Dear editor and authors,

I have read the manuscript "Water-gas two-phase flow behavior of multi-fractured horizontal wells in shale gas reservoirs" with some interest. The authors develop and describe a model to describe the flow behaviour of water and gas in shale gas reservoirs. Although I have noted some points where the authors could perhaps improve their manuscript, pending these relatively minor changes I recommend the article for publication in Processes.

Yours faithfully,
Jamie I. Farquharson

Main comments:

In several places throughout the manuscript, the authors refer to oil, oil-wetting, or oil-water phases (for example lines 82, 252, 445, 453), despite the manuscript being about gas and water phases. There is unfortunately no attempt to validate the model with industry or experimental data, as such remaining entirely theoretical. If the authors were able to find and provide such data, the validity of their model would be greatly improved. Several semi-empirical relations are introduced throughout the manuscript (e.g. line 131, 149) without reference or justification. In each case, the authors should explain the derivation of the relation or provide a reference to another work. Moreover, it would be good to see some justification for the use of each model over alternatives in each case. In most examples, the system is still evolving at t = tmax (see Figure 3, for example, where the water rate never plateaus), indicating that the simulations were terminated before the system achieved a steady-state. The authors should mention this explicitly in the text, or re-run the models with a longer timeframe. The model assumes a single fracture event. Is this a repeatable process? What would the be the effect of introducing a second (or third or fourth) hydrofracture event?

Minor comments:

The caption for Figure 4 refers to the gas distribution after 100 days, but in the text (e.g. line 349) it is stated that the figure shows distribution after 1 day. Which of these is correct?

There are numerous minor grammatical and typographical errors throughout the manuscript, some of which I have listed here:

Line 32 and 34: No need for "The" before "fluid flow"

line 35: "and the" should be "but"

line 68: Replace "The scholars..." with "Those scholars..."

Line 82: "describes"

Line 86: "flow models"

Line 87: "fractures"

Line 88: "networks"

Line 88 and line 233: "Replace "analyzed" with "described"

Line 106: Replace "is" with "has been"

Line 146: Replace "the diameter of gas molecular" with "...the diameter of the gas molecule"

Line 169: Remove "of"

Line 317: Replace "is required to be verified" with "requires verification"
